# Evaluation of Solar-Driven Photocatalytic Activity of Thermal Treated TiO_2_ under Various Atmospheres

**DOI:** 10.3390/nano9020163

**Published:** 2019-01-29

**Authors:** Reza Katal, Saeideh Kholghi Eshkalak, Saeid Masudy-panah, Mohammadreza Kosari, Mohsen Saeedikhani, Mehrdad Zarinejad, Seeram Ramakrishna

**Affiliations:** 1Department of Civil & Environmental Engineering, National University of Singapore, Singapore 117576, Singapore; reza.katal@u.nus.edu; 2Department of Mechanical Engineering, Center for Nanofibers and Nanotechnology, National University of Singapore, Singapore 117575, Singapore; s.kholghi143@gmail.com; 3Department of Electrical and Computer Engineering, National University of Singapore, Singapore 119260, Singapore; saeid1@e.ntu.edu.sg; 4Department of Chemical and Biomolecular Engineering, Faculty of Engineering, National University of Singapore, Singapore 119260, Singapore; kosari@u.nus.edu; 5Department of Materials Science and Engineering, National University of Singapore, Singapore 117583, Singapore; saeedikhani@u.nus.edu; 6Singapore Institute of Manufacturing Technology (SIMTech), A*STAR (Agency for Science, Technology and Research), Singapore 138634, Singapore

**Keywords:** oxygen vacancy, thermal treatment, vacuum, hydrogen, degradation

## Abstract

In this report, the photocatalytic activity of P25 has been explored and the influence of thermal treatment under various atmospheres (air, vacuum and hydrogen) were discussed. The samples’ characteristics were disclosed by means of various instruments including X-ray diffraction (XRD), Electron paramagnetic resonance (EPR), X-ray photoelectron spectroscopy (XPS) and UV–vis. This study also accentuates various states of the oxygen vacancy density formed inside the samples as well as the colour turning observed in treated P25 under various atmospheres. Produced coloured TiO_2_ samples were then exploited for their photocatalytic capability concerning photodegradation of methylene blue (MB) using air mass (AM) 1.5 G solar light irradiation. Our findings revealed that exceptional photocatalytic activity of P25 is related to the thermal treatment. Neither oxygen vacancy formation nor photocatalytic activity enhancement was observed in the air-treated sample. H_2_-treated samples have shown better photoactivity which even could be further improved by optimizing treatment conditions to achieve the advantages of the positive role of oxygen vacancy (O-vacancy at higher concentration than optimum acts as electron trapping sites). The chemical structure and stability of the samples were also studied. There was no sign of deteriorating of O_2_-vacancies inside the samples after 6 months. High stability of thermal treated samples in terms of both long and short-term time intervals is another significant feature of the produced photocatalyst.

## 1. Introduction

The intermediate conducting ability of titanium dioxide (TiO_2_) has converted it to an oxide semiconductor which holds an especial place in environmental applications that require a photoactive material; TiO_2_ only functions with the aid of UV light due to its large energy band gap [1,2,3]. Since pure TiO_2_ cannot absorb the wavelengths greater than 387 nm and solar irradiation’s wavelength lies beyond that margin, thus, solar visible light cannot be applicable. Majority of studies with respect to band gap engineering are focused to improve TiO_2_ photocatalytic performance. In this respect, one way is to decorate the TiO_2_ surface by dopants, such as metals and non-metals. Cations such as Mn, Ag, Au, Cr, La and Fe were utilized as metal dopants and they could decrease TiO_2_ bandgap when are completely dispersed into its lattice [4,5,6]. Non-metal dopants such as N [7,8,9,10], C [11,12,13] and S [13], was widely applied to enhance the TiO_2_ light absorption. A mixed composition of both metals (e.g. transition metals) and non-metals can also be used for doping, co-doping and even tri-doping of TiO_2_ surface for turning down the TiO_2_ band gap [11,14,15,16,17,18]. The very first report that demonstrated defect-engineered TiO_2_ can be generated by thermal treatment was reported by Chen et al. in 2011 [19]. They found that thermal treatment under high-pressure hydrogen atmosphere (200 °C for 5 days) leads to the generation of black TiO_2_ with improved photoactivity. The generated defects alter the band gap value and optical properties of TiO_2_ which in turn boosts photocatalytic activity [19]. It was also reported that Ti^3+^ self-doped TiO_2_ possesses considerable visible light absorption and noticeable photocatalytic activity [20,21,22,23,24,25]. Different strategies are recently used to synthesize Ti^3+^ doped TiO_2_ (generation of oxygen vacancy) including plasma treatment, reducing treatment in situ, calcination in reducing conditions and ion beam enhanced deposition [20,26,27,28,29,30,31]. Due to processes simplicity and low-cost facilities, reducing treatment in situ and calcination in reducing conditions and/or atmospheres have attracted more attention [32,33,34]. Kong et al. reported an achievement to higher photocatalytic activity for TiO_2_ as a result of raising the relative concentration ratio of surface to bulk defects [35]. In another investigation, Yan et al. found that the presence of oxygen vacancies on the surface of TiO_2_ samples could result in a superior photocatalytic activity through the reaction thanks to the well-segregation of charge carriers [36]. Moreover, Kaifu et al. elucidated that the hydrogenated TiO_2_ displayed higher photoactivity in comparison with the pure anatase TiO_2_ under simulated sunlight irradiation. This phenomenon was ascribed to narrower bandgap of hydrogenated anatase TiO_2_ (visible-light absorption) because of the Ti^3+^ formation in frameworks and surface disorders [37]. It was also reported that TiO_2_ can be used under visible light irradiation if it is treated thermally under vacuum atmosphere which generates free O_2_-vacancy states and Ti^3+^ [38]. A report by Pei et al. states that the TiO_2−x_ showed an improvement in hydrogen evaluation rate and moreover, the authors demonstrated that generated defects by trapping electrons could suppress the reacting of photogenerated e^−^–h^+^ pairs [39].

In this work, generation of O_2_-vacancies in P25 samples was triggered by post-treatment under vacuum and hydrogen atmospheres at different temperatures. Coloured TiO_2_ synthetic procedure from P25 powder was described in detail. The influencing parameters on photocatalytic performance including optical properties, the oxygen vacancies and Ti^3+^ in Coloured TiO_2_ samples have been examined. The photodegradation experiments were performed with a special attention to the effect of temperature on the features of treated P25 powder such as crystal size and band gap. A comparison is made between the oxygen vacancies generated inside the samples and the observed colour turning in vacuum-treated and H_2_-treated samples. Lastly, produced samples were subjected to a degradation reaction under solar light illumination in which they were placed in an aqueous solution containing methylene blue (MB) to analyse their photocatalytic degradation capability. In this study, X-ray photoelectron spectroscopy (XPS) was used to analyse the bonding of Ti and O atoms in the samples. For the first time, we found that the Ti 2p peaks of V-400 shifted to higher binding energies, whereas peaks of H-400 shifted to lower binding energy. A new strategy has been used for detection of free oxygen vacancy by EPR analysis. Generally, at sufficient vacancy concentration, a free oxygen vacancy band also can be formed below the conduction band. To evaluate the existence of free oxygen vacancies, H-400 and V-400 samples were treated under 1.5 G solar light illumination for 30 min. Two key features of the as-produced photocatalyst, that is, stability and reusability, especially in terms of industrial scale application, were also checked in this investigation. We found that V-400 showed better stability for photocatalytic degradation of MB than H-400 after 5 cycles.

## 2. Materials and Methods

### 2.1. Instruments

The X-ray diffractometry (XRD) of samples was performed by Cu–Kα radiation by Shimadzu, lab-X XRD-6000 X-ray Diffraction instrument (Kyoto, Japan). Transmission electron microscopy (TEM, JEM-2010, Tokyo, Japan) was used for the detection of formed disordered layer in the samples. The elemental composition (chemical state and electronic state of the elements) of the surface compositions on the samples identification were performed by X-ray photoelectron spectroscopy (XPS) using Kratos AXIS UltraDLD (Kratos Analytical Ltd., Manchester, UK). The ultraviolet-visible spectrophotometer (UV-vis, Shimadzu, UV-3600, Kyoto, Japan) was utilized for ultraviolet-visible diffuse reflectance spectra in the range of 200–800 nm with BaSO_4_ as the reference. The sample band gap (*E*_g_) was calculated by Equation (1):*E*_g_ = hc/λ(1)
where h and c are Planck’s constant (6.626 × 10^−34^ J·s) and speed of light (3 × 10^8^ m/s) respectively; λ is the extrapolated wavelength (nm) at which the absorbance value reaches the instrument limit. Electron paramagnetic resonance (EPR) spectra (JEOL FA200 SPECTROMETER (X-Band with LN2 VT accessories) (Pleasanton, CA, USA) were used at room temperature, at a microwave frequency of 9.85 GHz for confirmation of high spin Ti^3+^ as well as oxygen vacancy.

### 2.2. Sample Preparation

Commercial P25 (Aeroxide P25 nanopowder (with average particle size of 21 nm), ≥99.5% trace metals basis, 718467 Sigma-Aldrich, St. Louis, MO, USA) was utilized as TiO_2_ source and other samples were prepared from P25.

Sample A-400: 0.5 g of P25 was placed into an alumina crucible and calcined at 400 °C in a muffle furnace under air condition for 4 h.

Sample H-400: P25 were placed in a quartz tube and heated in a tube furnace under a 50 sccm gas flow of pure hydrogen for 4 h at 400 °C. After hydrogen treatment, for stabilization issue, sample H-400 was kept in vacuum for 2 h.

Sample V-400: This sample was prepared as sample A-400 but treated under vacuum condition.

### 2.3. Photocatalytic Degradation Experiments

Methylene blue (MB) was applied as an organic pollutant to test the shape-dependent photocatalytic activity of anatase TiO_2_. In a typical photocatalytic degradation experiments, 10 mg of the photocatalyst was suspended in 30 mL of a 10 ppm MB aqueous solution under atmospheric air. A xenon arc lamp (150 W) was used for AM 1.5 G solar light illumination. After photocatalytic degradation experiments in a required time, determined volume of each experiment were sampled and their concentrations were analysed by an UV-vis spectrophotometer. In order to ensure the establishment of an adsorption/desorption equilibrium among photocatalysts was magnetically stirred in the dark for 30 min. The value of *E*_g_ was calculated by plotting (αhυ)^2^ versus hυ and extrapolating the plot to (αhυ)^2^ = 0.

The Pseudo-first-order plots of the photocatalytic degradation was used to describe the kinetic of MB photocatalytic degradation by samples under 1.5 G solar light illumination:ln(*C*_t_/*C*_0_) = −*K*_app_*t*(2)
where, *C*_0_ (ppm), *C_t_* (ppm) and *K*_app_ (min^−1^) are MB initial (*t* = 0), final concentration (at time *t*) in aqueous solution and Pseudo-first-order constant, respectively.

The degradation (photodegradation) efficiency was calculated by Equation (3):*Degradation efficiency* (%) = ((*C*_0_ − *C_t_*)/*C*_0_) × 100(3)

The recycling of the photocatalyst was implemented as follows: after a first photodegradation cycle, the treated solution of the dye was centrifuged with a rotation of 10,000 rpm for 10 min to remove photocatalyst. The liquid phase was filtered by a vacuum system with a Millipore membrane (0.45 μm, Merck, Darmstadt, Germany) and the solid phase containing the photocatalyst was carefully separated for reuse. After allowing it to dry in an oven for 12 h at 50 °C, the separated photocatalyst was added again to the next cycle. The process was repeated 5 times. The produced CO_2_ in gas phase was monitored by continuous analysers, measuring CO, CO_2_ (Uras 26, ABB) gaseous concentrations.

## 3. Results and Discussion

### 3.1. Physiochemical Characterization of Photocatalyst

XRD diffractographs of four samples including P25, A-400, V-400 and H-400 was shown in Figure 1. The patterns indicate that the samples have the same crystalline structure (including anatase and rutile) after thermal treatment. Increasing the intensity of both anatase and rutile peaks indicates an improvement of crystallinity of samples. As can be seen, after thermal treatment, the peak width of all samples were close together which is indicative of similar crystal size of all samples. The SEM of P25 and thermal treated sample (H-400) are shown in Appendix A. As can be seen, by thermal treating of samples, an increment in size of samples was observed. The surface area of P25 and thermal treated samples is presented in Appendix A.

Another piece of information to elucidate the chemical structure of samples is provided by showing the spectrum of UV-vis diffuse reflection of four samples in Figure 2a. As is shown in this figure, the light absorption characteristics of P25 is greatly affected in both H-400 and V-400 samples which is due to oxygen vacancy generation. A sharp increase in the absorbance of all samples can be observed ranging from 400 to 200 nm, which is ascribed to anatase phase of TiO_2_. The H-400 and V-400 samples show a redshift in absorption edge of UV-vis range. Measured by given Equation (1), the band gap values of three samples are listed in Figure 2b along with the physical appearance of powders. According to the calculated values, they can be arranged from the lowest to highest band gap in the order of H-400 < V-400 < A-400 < P25. The band gap value of H-400 and V-400 samples are smaller than that of P25, resulting in its light-absorption toward redshift. The colour turning of samples is illustrated in Figure 2b. The white colour of P25 turned to grey after thermal treatment under hydrogen and vacuum condition; although, the colour of H-400 was slightly darker grey. This colour turning is ascribed to generation of oxygen vacancy in H-400 and V-400. Meanwhile, no colour turning in A-400 was detected. Appendix A shows the plot of (αhυ)^2^ versus hυ for H-400, V-400 and P25. The calculated *E*_g_ values are shown in Figure 2b.

The study of surface chemical composition of thermally-treated samples (H-400 and V-400) and P25 was accomplished utilizing X-ray photoelectron spectroscopy (XPS). Figure 3 gives the analysed high resolution XPS spectra of bonding of titanium (Ti) and oxygen (O) atoms. Revealed by XPS spectrum of Ti in Figure 3a, the Ti 2p spectrum has two peaks at 459.0 and 464.7 eV, ascribed to the binding energy of 2p_3/2_ and 2p_1/2_ of TiO_2_, which are a common hint of the Ti–O–Ti bonds in TiO_2_ samples [40]. The respective Ti 2p peaks shifted to higher binding energies for V-400 sample while the peaks of H-400 shifted to lower ones. This evidence is indicating that Ti^3+^ ions are possibly formed in both H-400 and V-400 samples [41]. Moreover, the O 1s binding energy of H-400 and V-400 (Figure 3b) also showed a similar trend resembling to that of Ti 2p spectrum. These results are indicative of the influence of Ti^3+^ formation in the bond interaction states of Ti–O bond [42].

As mentioned, after H_2_ treatment, the Ti 2p peaks of the H-400 shifted to the lower values of binding energies, indicating that oxygen vacancies (Ti^3+^ sites) were generated in H-400 sample during hydrogenation [43,44,45]. According to literature, there are other reports that relate this peak shifting to low-energy values to the Ti–H bonds formation on the surface of samples [43,46,47]. Thermal treatment under vacuum or hydrogen yields a reduced atmosphere, which consequently trigger the formation of Ti^3+^ ions and oxygen vacancy on the surface of the samples (partial reduction of TiO_2_ under reduced conditions) [43]. Thermal treatment under vacuum caused Ti 2p and O 1s photoelectron XPS spectra to shift to higher binding energies. One may identify this phenomenon as a sign of generation of neighbouring oxygen vacancies which indicates a high electron-attracting affinity [48]. In order to test for Ti^3+^ and oxygen vacancies, Ti 2p and O 1s XPS spectra are measured and the Ti 2p_3/2_ peak of the vacuum activated sample became unsymmetrical compared with the peak of pure P25 (Figure 2b), indicating the existence of Ti^3+^ [49]. Furthermore, the Δ*E* value between Ti 2p_1/2_ and Ti 2p_3/2_ was 6.2 eV for the vacuum activated sample, suggesting the presence of Ti^3+^ [37,50,51].

The investigation over revealing the chemical properties of samples were continued by performing EPR. This method was used to study unpaired electrons in paramagnetic samples, which can be described as one of the powerful strategy to detect oxygen vacancies and Ti^3+^. Figure 4 demonstrates the EPR spectra of P25, H-400 and V-400 determined at room temperature. No signal was observed for P25 and A-400 (EPR spectra of A-400 was not presented), indicative of no paramagnetic species presence in these samples. On the contrary, obvious signals at g equals to 2.001 in the EPR spectra of H-400 and V-400 might be ascribed to oxygen vacancy generation in these samples [52,53]. It must be mentioned that surface oxygen vacancies cannot be directly analysed by EPR; generally, the surface oxygen vacancies possess two electrons, where Ti^4+^ will be reduced to Ti^3+^ by some of them. Therefore, the Ti^3+^ ions generation tacitly corroborated that oxygen vacancies are present in the samples [42].

Some studies have indicated the feasibility of synthesizing visible-light or sunlight responsive TiO_2_ using generating Ti^3+^ accompanied by high vacancy concentration; it was reported that higher vacant sites can develop the oxygen vacancy states beneath the conduction band [54,55]. Additionally, the formation of free oxygen vacancy band below conduction band is also possible if enough vacancy concentration existed [55,56]. To measure the amount of formed free O_2_-vacancies, H-400 and V-400 samples were treated under 1.5 G solar light illumination for 30 min. As shown in Figure 5, after the illumination treatment, an increment in the width and intensity of the EPR single peak was observed. According to this figure, there is a shift to higher values of “*g*” for the EPR single peak. One may infer that this shift indicates the existence of free O_2_-vacancies resulted from trapped photo-excited electrons [55,57].

### 3.2. Solar light-driven Photocatalytic Degradation of Methylene Blue (MB) in Aqueous Solution

The MB degradation reactions in aqueous solution using the thermally treated samples were executed to probe the photocatalyst efficiency under solar light illumination. The oxygen vacancy concentration of samples was controlled by adjusting the temperature of thermal treatment for both conditions (i.e. hydrogen atmosphere and vacuum). The calcined TiO_2_ precursor (i.e. P25) at 200, 300, 400 and 500 °C under hydrogen atmosphere (namely, H-200 to H-500) were applied to degrade MB in an aqueous medium placed below the 1.5 G solar light irradiation. As shown in Appendix A, by increasing the calcination temperatures until 400 °C, the photocatalytic degradation rate increased and H-400 showed the highest degradation; for H-500, a slight reduction in MB photodegradation rate was observed. The same result was also acquired by the samples treated under vacuum (i.e. V-400), as shown in Appendix A. It is worth noting that the surface O_2_-vacancies can separate the photo-excited electron-hole pairs and significantly enhance the photocatalyst activity [58,59]. The noticeable photocatalytic activity of both H-400 and V-400 samples can be attributed to their narrowed band gap, charge separation and charge transfer owing to presence of O_2_-vacancies [60]. The H-500 and V-500 displayed the lower photocatalytic activity than that of H-400 and V-400. The oxygen vacancy acts as an ionized donor atom of positive charge; at higher oxygen vacancies density than optimum, more oxygen vacancies generation is taken place near the surface, thus, higher amount of positive charges is accumulated on the surface of photocatalyst. The photogenerated electrons and positive charges can be more easily recombined together; generating oxygen vacancies has moreover high potential to recombine with photoinduced holes [60,61,62,63,64]. This process may pose negative impact on the photocatalytic performance of coloured TiO_2_ sample and might be the chief cause of low photocatalytic activity of both H-500 and V-500 samples.

Figure 6 illustrates the concentration profile of MB versus time during the photocatalytic reaction under 1.5 G solar light irradiation. Using the kinetics data presented in this figure, the apparent rate constants (*K*_app_) of MB degradation were 0.0096, 0.0434 and 0.0544 min^−1^ for P25, V-400 and H-400 respectively. Although the excitation of P25 is possible by UV light but in the absence of the surface state to delay the recombination, its photocatalytic performance is low. As can be seen, H-400 and V-400 (containing oxygen vacancies) exhibit higher photodegradation rate than the P25 and A-400. As mentioned above, the oxygen vacancies in H-400 and V-400 can act as the separating hubs for the photoexcited electron/hole pair, thus this is the reason why they showed better photocatalytic performance than P25 and A-400 [1,52]. The comparison between photocatalytic degradation rate of H-400 and V-400 indicates the thermal treatment under hydrogen atmosphere leads to higher photocatalytic activity and efficiency. As can be seen, the concentration of MB does not change during the reaction without photocatalyst under 1.5 G solar light illumination. Under dark condition in presence of photocatalyst, less than 10% of MB was removed due to the adsorption of MB by photocatalyst.

To remove the lattice oxygen (O_L_) in the form of water molecules and to produce an O-vacancy (V_O_), the H_2_ molecules will combine with the O_L_ that are present in the TiO_2_ lattice under the thermal treatment [58]. As a result, two electrons will be produced that should be accommodated at the vacant site [55].
H_2_ + O_L_ → H_2_O + V_O_ + e^−^(4)

On the contrary, oxygen removal from TiO_2_ under vacuum atmosphere could be described Equation (5):O_L_ = 1/2 O_2_ (g) + V_O_ + e^−^(5)

Equation (5) demonstrates that oxygen vacancies will also being produced by thermal treatments under vacuum atmospheres, which was reported by previous studies [55,58]. Therefore, although H-400 and V-400 samples have received the same thermal treatment in terms of time and temperature, there exist a variation in their photocatalytic activity which might be due to the difference in O-vacancy production mechanism. Moreover, thermal treatment under hydrogen atmosphere was found to be more effective for oxygen vacancy production [55,58].

In order to elucidate a better picture of the photocatalyst (P25, V-400 and H-400) properties, prepared in this study, the photoluminescence (PL) spectra were obtained at excitation wavelength of 325 nm and presented in Figure 7. From this result, two major peaks at ca. 466 and 548 nm can be identified in the PL spectrum of P25. To a great degree, one may decipher the reduction in PL intensity into existence of higher electron–hole segregation (long-lasting photogenerated charge carriers) [59,60]. The PL intensity of H-400 and V-400 was lower than P25 which is indicative of H-400 and V-400 facilitating reduced recombination rate of photogenerated electron-hole pairs [58]. Therefore, the generation of the oxygen vacancy on the surface of H-400 and V-400, which is behaving as electron scavengers, suppress photoluminescence intensity. This is why H-400 and V-400 showed higher photocatalytic performance than P25 [60].

In order to better understand the enhancement of visible light absorption, the photocatalytic degradation tests has been performed for the samples under visible light (λ > 420 nm) irradiation (Appendix A). As can be seen, P25 does not show any special activity under visible light irradiation; other two samples show the higher activity. This clearly shows that the role of visible light absorption improvement on the photocatalytic activity of samples.

### 3.3. The Influence of Free Radicals on MB Photocatalytic Degradation and Mechanistic Studies

In a typical photocatalytic degradation reaction, the superoxide radical (O_2_·^−^), hydroxyl radical (OH·) and hole (h^+^) can be described as active species in process [62,63]. To evaluate their role in MB photodegradation under 1.5 G solar light illumination, a series of scavenger tests was done on V-400 and H-400 samples. In order to carefully examine the role of h^+^, OH· and O_2_·^−^ radicals, the degradation reaction was conducted with and without scavenger such as ammonium oxalate (AO), ethanol and sodium azide (SA) scavengers [62,63]. According to Figure 8, addition of aforementioned scavengers to the reaction resulted in decreasing MB degradation efficiency by the photocatalysts. This finding validates the significant contribution of active sites on the photocatalyst performance. Our findings proved that h^+^ radicals play key role in MB photodegradation. Majority of reactive species in the MB degradation reaction can be OH· radicals. It is believed that photo-generated holes are responsible for OH· radicals generation and direct oxidation of pollutants in this photocatalytic process. In the case of O_2_·^−^ radicals, transfer of photo-excited electrons from V_O_ states to TiO_2_ conduction band where they react to oxygen molecules is considered as key pathway for O_2_·^−^ radicals generation. It is worth noting that instability of the superoxide free radicals in the solution causes them to be converted into OH· radicals [64,65]. The bandgap narrowing of V-400 and H-400, provides a suitable condition for absorption of light in a wider range in comparison to P25. However, the charge separation can be counted as one of the crucial parameters influencing the photocatalytic efficiency of methylene blue degradation.

Under illumination, charges are separated, h^+^, O_2_·^−^, OH· radicals are generated which have high potential to degrade MB into carbon dioxide and other inorganic compounds [65,66,67,68]. Below, we listed several pathways that are likely to occur during photocatalytic decomposition of MB:O_2_+ e^−^ → O_2_·^−^(6)
O_2_·^−^ + 2H^+^ + e^−^ → H_2_O_2_(7)
H_2_O_2_+ e^−^ → OH· + OH^−^(8)
H_2_O → H^+^ + OH^−^(9)
OH^−^ + h^+^ → OH·(10)
h^+^ + MB → CO_2_ + products*(11)
(O_2_·^−^, OH·) + MB →water + CO_2_ + products*(12)

Appendix A shows the concentration of variation of CO_2_ produced during the photocatalytic degradation of MB by using samples as a photocatalyst under simulated sunlight irradiation. It could be clearly seen from Appendix A that the concentration of CO_2_ gradually increased along the light irradiation time.

The electron transfer between anatase and rutile phases in V-400 and H-400 samples is shown in Figure 9. As shown, oxygen vacancy states act as key parameter in improving of photocatalytic activity of V-400 and H-400 under 1.5 G solar light illumination [37].

### 3.4. Stability and Recyclability of The Photocatalysts

Stability and good recyclability are two essential features for industrial scale application of photocatalysts [69,70]. The stability and recyclability of photocatalyst (V-400 and H-400) towards MB photodegradation were tested for five consecutive runs and the obtained data are presented in Figure 10. From this figure, it is clear that there only exists a negligible loss (< 10%) in photodegradation efficiency after 5 successive cycles. Meanwhile, V-400 showed slightly better stability than H-400 after 5 cycles. The photodegradation efficiency of MB by H-400 and V-400 using 1.5 G solar light illumination at different time storage is given in Figure 11. From this standpoint, no significant decline of the photodegradation efficiency for MB was also noticed. This result certifies the appropriate stability of photocatalysts for long-term applications. To find the reason for the photocatalytic activity reduction, EPR analysis of the samples was used. The results are shown in Appendix A. The EPR signal intensity shows the oxygen vacancy density; as can be seen, the signal intensity of the used samples is slightly reduced. It can be concluded that the photocatalytic activity of used samples is reduced based on the reduction of oxygen vacancy density in used samples. UV-Vis spectra and images of P25, V-400 and H-400 after 6 months storage are presented in Appendix A. As shown, there is almost no trace of colour change in the samples and UV-Vis spectra of samples slightly reduced (less than 10%). These indicated acceptable stability of photocatalyst for long-term storage.

## 4. Conclusions

In the present work, generation of oxygen vacancy in P25 samples by thermal treatment under vacuum and hydrogen atmospheres at different temperatures (200, 300, 400 and 500 °C) was successful. The mechanistic studies on the preparation of Coloured TiO_2_ from P25 powder was also carried out. The samples H-400 and V-400 show redshift in absorption edge of UV-visible range. The band gap value of H-400 and V-400 are narrower than that of P25. We noticed colour turning of P25 from white to grey after being treated thermally under hydrogen atmosphere (H-400) or vacuum condition (V-400); although, the colour of H-400 was slightly darker grey. The investigation over revealing the chemical properties of samples were continued by performing EPR, which was used to study unpaired electrons in paramagnetic samples. No signal was observed for P25 and A-400, demonstrating no paramagnetic species exist in these samples. On the contrary, obvious signals at g equals to 2.001 in the EPR spectra of H-400 and V-400 might be ascribed to oxygen vacancy generation in these samples. The P25, H-400 and V-400 were exploited as photocatalyst in the degradation reaction of methylene blue (MB) in the solution under AM 1.5 G solar light illumination; it was found that thermal treatment considerably improved the photocatalytic activity of P25. Hydrogen treated samples displayed better photoactivity. Optimum treatment conditions for different atmospheres were established to achieve the advantages of the positive role of oxygen vacancy; it was found that oxygen vacancy at higher concentration than optimum act as electron trapping sites. The PL spectra obtained at excitation wavelength of 325 nm for all photocatalysts (H-400, V-400 and P25) showed lower intensity for H-400 and V-400 samples than that of P25. This is an indication of the fact that H-400 and V-400 facilitate reduced recombination rate of photogenerated electron-hole pairs. The stability of generated O-vacancy density on the surface of the photocatalysts was tested after elapsing six months. It was found that there was no significant change in density. The stability of the samples in an aqueous environment was also examined. The result certified the appropriate stability of photocatalysts for long-term applications. Potentially, our thermally-treated TiO_2_ samples can be used for photocatalytic applications and processes in large scales.

## Figures and Tables

**Figure 1 nanomaterials-09-00163-f001:**
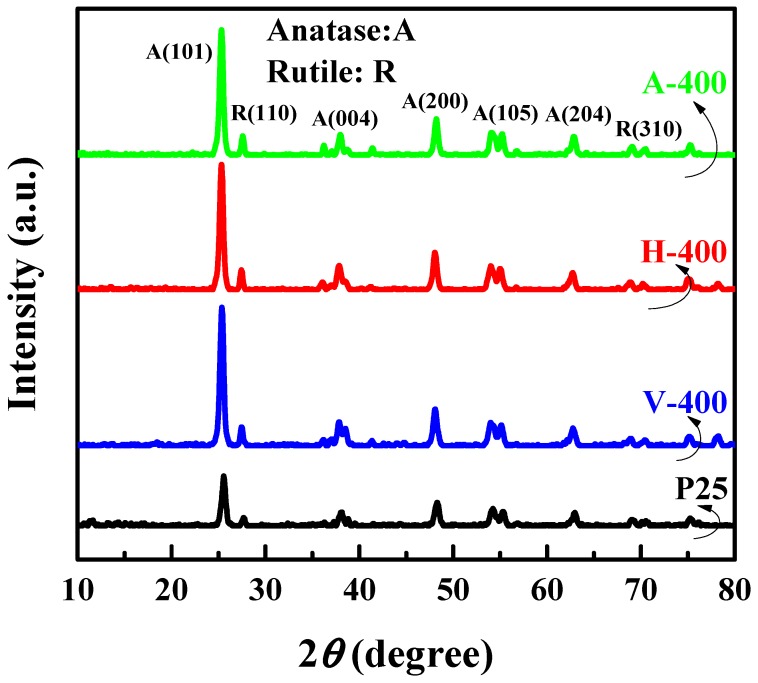
X-ray diffraction (XRD) pattern of P25, A-400, V-400 and H-400.

**Figure 2 nanomaterials-09-00163-f002:**
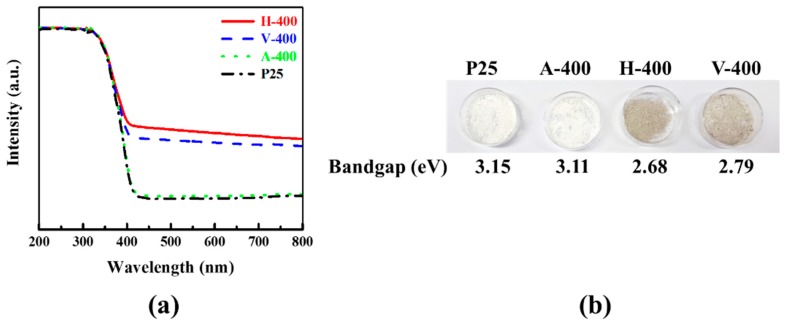
(**a**) UV-vis spectra, (**b**) colour turning image and energy band gap of samples.

**Figure 3 nanomaterials-09-00163-f003:**
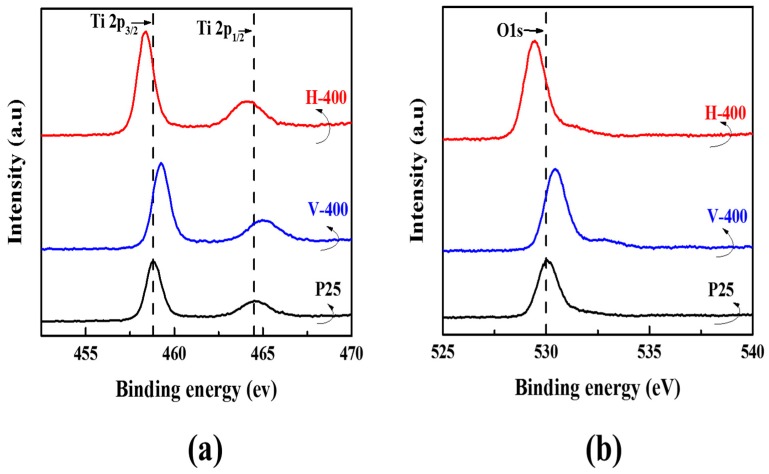
XPS spectra of Ti 2p peaks (**a**) and O 1s peaks (**b**) for P25, H-400 and V-400.

**Figure 4 nanomaterials-09-00163-f004:**
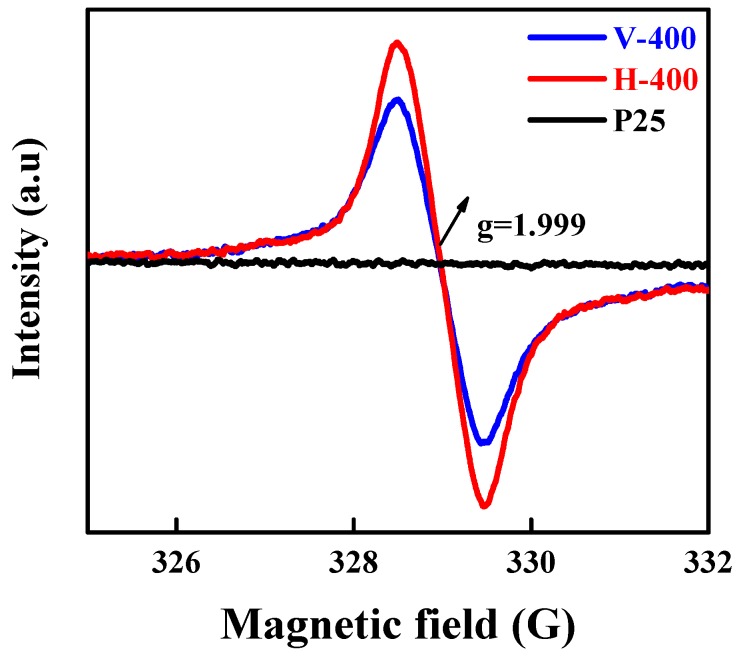
EPR spectra recorded at room temperature for P25, H-400 and V-400.

**Figure 5 nanomaterials-09-00163-f005:**
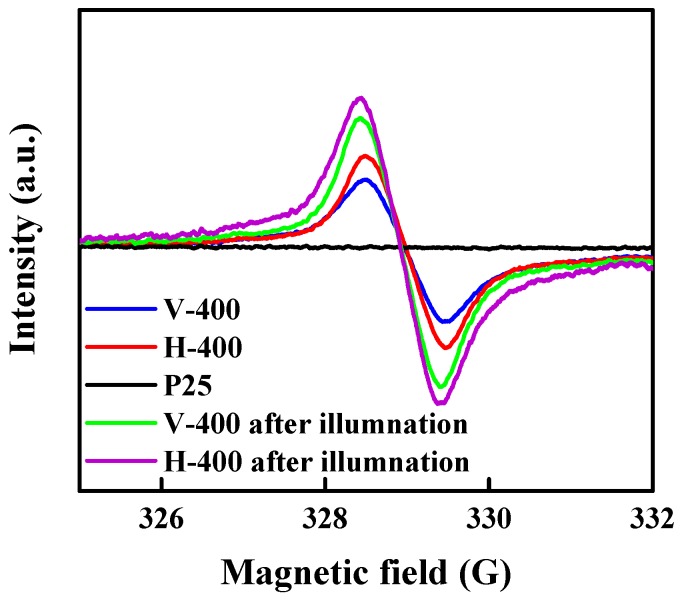
EPR spectra for P25, H-400, V-400 after and before illumination under 1.5 G solar light for 30 min.

**Figure 6 nanomaterials-09-00163-f006:**
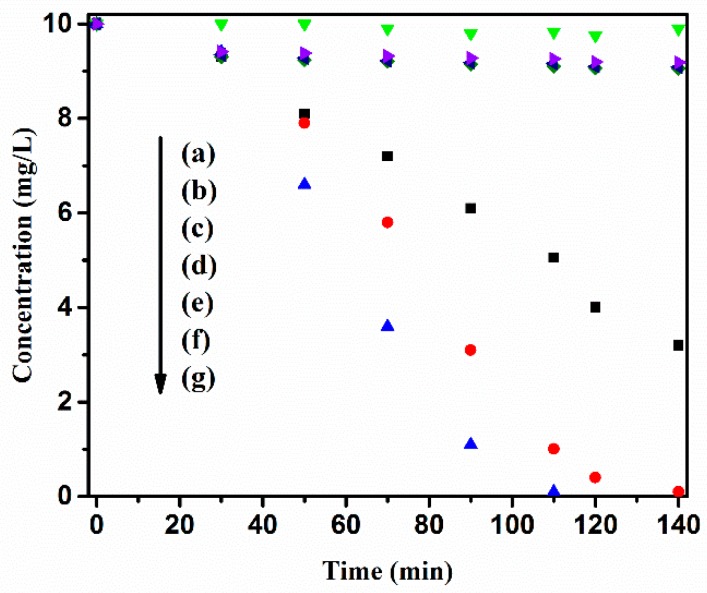
Time profiles of photocatalytic degradation of MB for (**a**) without photocatalyst; (**b**) H-400; (**c**) V-400; (**d**) P25 without irradiation; (**e**) P25; (**f**) V-400; and (**g**) H-400 under 1.5 G solar light illumination.

**Figure 7 nanomaterials-09-00163-f007:**
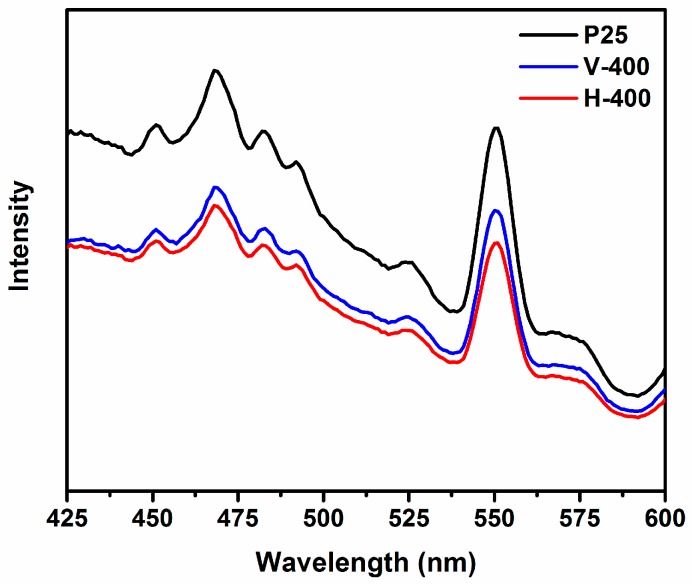
PL spectra of P25, V-400 and H-400 (excitation wavelength was 325 nm).

**Figure 8 nanomaterials-09-00163-f008:**
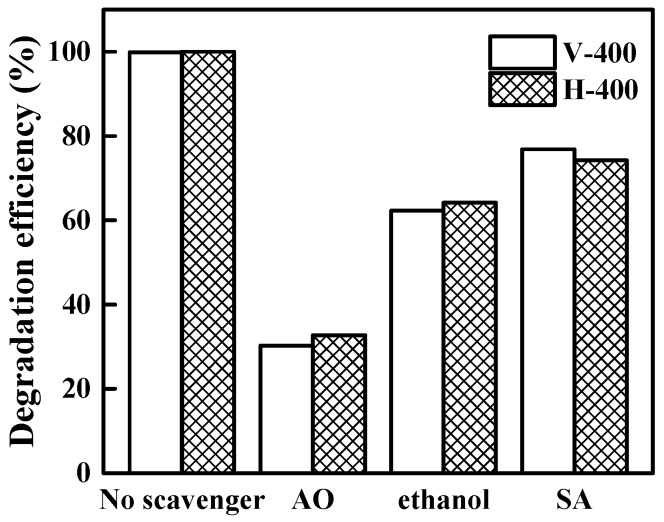
Degradation (%) of MB in without quencher and in presence of scavengers.

**Figure 9 nanomaterials-09-00163-f009:**
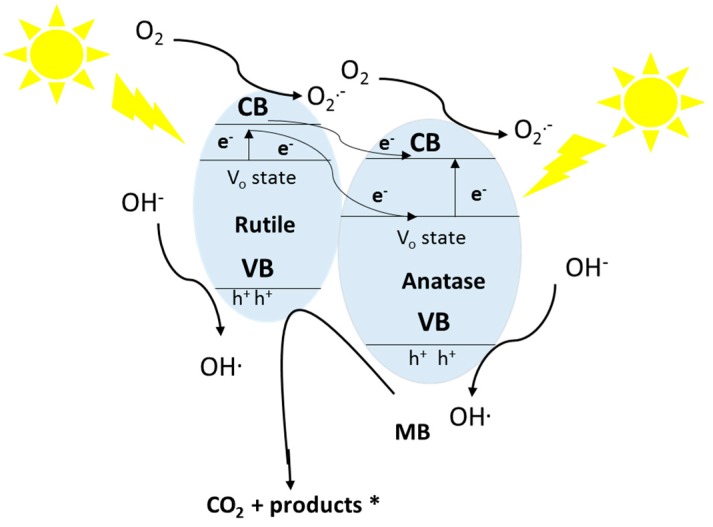
The photo-excited electrons-transfer under the simulated sun light irradiation for V-400 and H-400.

**Figure 10 nanomaterials-09-00163-f010:**
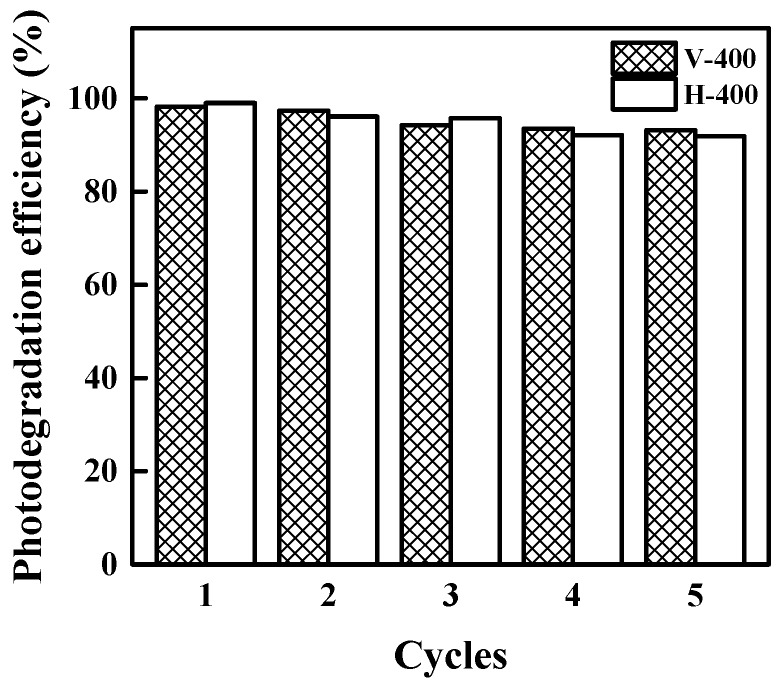
MB degradation efficiency (%) by H-400 and V-400 under AM 1.5 G solar light irradiation at different cycles.

**Figure 11 nanomaterials-09-00163-f011:**
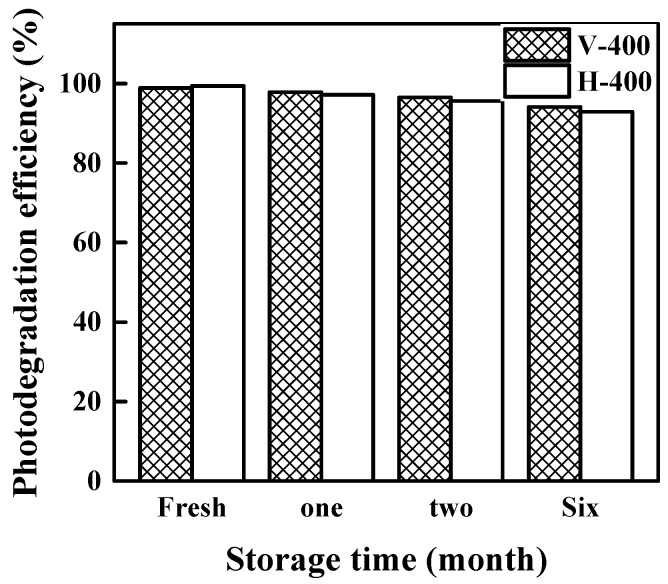
MB degradation efficiency by V-400 and H-400 with different storage time under AM 1.5 G solar light irradiation.

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
