# Peer review of "Evaluation of Solar-Driven Photocatalytic Activity of Thermal Treated TiO2 under Various Atmospheres"

_nanomaterials, 2019, doi:10.3390/nano9020163_

Round 1
Reviewer 1 Report
Manuscript Number: Nanomaterials
Manuscript Title:” Evaluation of solar-driven photocatalytic activity of thermal treated TiO2 under various atmospheres”.
Recommendation: Accept after minor revision
Additional comments: The target of the manuscript is evaluation of thermal treated TiO2 photocatalysts for water purification in order to establish essential factors that lead to increase the photocatalytic activity. This target is important and actual. So, the manuscript is dedicated to an actual problem of chemistry, water purification technologies. However, there are some problems in the manuscript that should be corrected. 1. In line 91 is written: The sample band gap (Eg) was calculated by Eq. 3. However, there is not Eq. 3 in the manuscript. 2. On Figures 6, Kapp (min-1) is better to show separately in order to avoid confusion with right axes values. 3. Figure 7, If the intensities on the Figure are present in a.u. the relations between intensities should be relative. So, the highs of the peaks should be compared.
Author Response
The target of the manuscript is evaluation of thermal treated TiO2 photocatalysts for water purification in order to establish essential factors that lead to increase the photocatalytic activity. This target is important and actual. So, the manuscript is dedicated to an actual problem of chemistry, water purification technologies.
However, there are some problems in the manuscript that should be corrected.
1. In line 91 is written: The sample band gap (Eg) was calculated by Eq. 3. However, there is not Eq. 3 in the manuscript.
· This mistake has been corrected.
2. On Figures 6, Kapp (min-1) is better to show separately in order to avoid confusion with right axes values.
· This comment has been amended in the manuscript.
3. Figure 7, If the intensities on the Figure are present in a.u. the relations between intensities should be relative. So, the highs of the peaks should be compared.
· This comment has been amended in the manuscript.

Reviewer 2 Report
The issue tackled in this manuscript is interesting per se for the scientific community in the field of photocatalysis. Often, many articles dealing with the pre-treatment TiO2 photocatalysts do not make such a systematic study, but instead assume that a particular pre-treatment is optimum for performance. Therefore, the approach in this work is acknowledged. Having said so, several drawbacks have been found during review, and to my opinion, should be amended. Characterisation of the materials needs to be thoroughly revised, and the supplementary information has to be provided (it is mentioned in the text yet is not available). The specific issues are outlined as follows:
1. The concept of oxygen vacancies is appropriately denoted as a most likely benefit for photocatalytic activity, especially under light of visible wavelengths, but this should then be checked for the materials under study by using UV-free or, even better, monochromatic light; photo-action studies would be the ideal proof for this. The authors should also clearly state whether they feel that it is the extension of band gap which is the cause of activity enhancement, or if instead, charge carrier separation is the main factor. Finally, the term”O2-vacancy” is not correct, since it is not dioxygen molecules which are absent from TiO2, yet individual O atoms.
2. TiO2 (P25) is not a microporous “molecular sieves” material, and mentions to this should be removed.
3. Refs. 1-3 are not appropriate since they are not general enough about the use of TiO2 photocatalysts, the following reviews are recommended: Chem. Rev., 2014, 114, 9919; Chem. Rev., 2014, 114, 9890; J. Photochem. Photobiol. C, 2012, 13, 3, 169.
4. For doped TiO2 materials, the following references should be taken into account. N-doped: Chem. Rev., 2014, 114, 9824; Cu-doped: Phys. Chem. Chem. Phys., 2014, 16, 3835; Appl. Catal. B-Environ., 2016, 180, 263.
5. It is said that black TiO2 is prepared, but the materials presented are not too similar to the original black TiO2 (Science, 2011, 331, 746), the reducing conditions employed in this manuscript are noticeably less severe.
6. The determination of band gaps should be graphically illustrated, since the explanation is not totally clear.
7. Photocatalytic experiments: was any wavelength filter used?
8. Photocatalytic experiments: how was contact of the suspension with atmospheric oxygen (as the oxidising agent) ensured?
9. Photocatalytic experiments: The procedure for reuse experiments should be thoroughly detailed.
10. XPS: The Ti signals are not “satellite” but genuine ones.
11. XPS: The lack of trend is suspicious; if both H- and V- samples have Ti3+ species, then both should show lowered Ti 2p binding energies, as also suggested by EPR; might this be due to the fact that the binding energies have not been corrected to adventitious carbon (284.6 eV)? Please explain in the experimental a section about energy correction in XPS.
12. PL spectra show a signal at 548 nm which the authors assign to Ti3+; then, why is this signal also seen for P25, which does not have Ti3+?
13. Degradation efficiency (Figs. 8 and 9) should be clearly defined.
14. Fig. 9: The conduction band of rutile is higher in energy according to Nat. Mater. 12 (2013) 798–801, and thus electrons are expected to flow from rutile to anatase.
15. The sentence in lines 299-300 must be corrected.
16. Which is the reason for the 10% decline in activity upon reuse (Fig. 10)? Maybe EPR, XPS or other techniques could be applied to the used photocatalysts to find the causes.
17. Refs. 8 and 19 should be completed/corrected.
Author Response
Reviewer 2:
The issue tackled in this manuscript is interesting per se for the scientific community in the field of photocatalysis. Often, many articles dealing with the pre-treatment TiO2 photocatalysts do not make such a systematic study, but instead assume that a particular pre-treatment is optimum for performance. Therefore, the approach in this work is acknowledged. Having said so, several drawbacks have been found during review, and to my opinion, should be amended. Characterisation of the materials needs to be thoroughly revised, and the supplementary information has to be provided (it is mentioned in the text yet is not available). The specific issues are outlined as follows:
1. The concept of oxygen vacancies is appropriately denoted as a most likely benefit for photocatalytic activity, especially under light of visible wavelengths, but this should then be checked for the materials under study by using UV-free or, even better, monochromatic light; photo-action studies would be the ideal proof for this. The authors should also clearly state whether they feel that it is the extension of band gap which is the cause of activity enhancement, or if instead, charge carrier separation is the main factor. Finally, the term”O2-vacancy” is not correct, since it is not dioxygen molecules which are absent from TiO2, yet individual O atoms.
· Thanks for the comment. Both of reduction in band gap value and improvement of charge carrier separation have positive effect on the photocatalytic activity of samples. Another piece of information to elucidate the chemical structure of samples is provided by showing the spectrum of UV–vis diffuse reflection of four samples in Fig. 2(a). As is shown in this figure, the light absorption characteristics of P25 is greatly affected in both H-400 and V-400 samples which is due to oxygen vacancy generation. A sharp increase in the absorbance of all samples can be observed ranging from 400 to 200 nm, which is ascribed to anatase phase of TiO2. The H-400 and V-400 samples show a redshift in absorption edge of UV−visible range. Measured by given Eq. 1, the band gap values of three samples are listed in Fig. 2(b) along with the physical appearance of powders. According to the calculated values, they can be arranged from the lowest to highest band gap in the order of H-400<V-400<A-400<P25. The band gap value of H-400 and V-400 samples are smaller than that of P25, resulting in its light-absorption toward redshift. In order to elucidate a better picture of the photocatalyst (P25, V-400 and H-400) properties, prepared in this study, the photoluminescence (PL) spectra were obtained at excitation wavelength of 325 nm and presented in Fig. 7. From this result, two major peaks at ca. 466 and 548 nm can be identified in the PL spectrum of P25. To a great degree, one may decipher the reduction in PL intensity into existence of higher electron–hole segregation (long-lasting photogenerated charge carriers) [56,57]. The PL intensity of H-400 and V-400 was lower than P25 which is indicative of H-400 and V-400 facilitating reduced recombination rate of photogenerated electron-hole pairs [58]. Therefore, the generation of the oxygen vacancy on the surface of H-400 and V-400, which is behaving as electron scavengers, suppress photoluminescence intensity. This is why H-400 and V-400 showed higher photocatalytic performance than P25 [57].
2. TiO2 (P25) is not a microporous “molecular sieves” material, and mentions to this should be removed.
· Thanks for the useful comments. The molecular sieve was wrongly used in this manuscript. Commercial P25 (Aeroxide P25 nanopowder (with average particle size of 21 nm)) was utilized as TiO2 source.
3. Refs. 1-3 are not appropriate since they are not general enough about the use of TiO2 photocatalysts, the following reviews are recommended: Chem. Rev., 2014, 114, 9919; Chem. Rev., 2014, 114, 9890; J. Photochem. Photobiol. C, 2012, 13, 3, 169.
· Thanks for the comments. These references has been added to the manuscript.
4. For doped TiO2 materials, the following references should be taken into account. N-doped: Chem. Rev., 2014, 114, 9824; Cu-doped: Phys. Chem. Chem. Phys., 2014, 16, 3835; Appl. Catal. B-Environ., 2016, 180, 263.
· Thanks for the comments. These references has been added to the manuscript.
5. It is said that black TiO2 is prepared, but the materials presented are not too similar to the original black TiO2 (Science, 2011, 331, 746), the reducing conditions employed in this manuscript are noticeably less severe.
· Thanks for the comments. The “black TiO2” used in the manuscript was not appropriate. The colored TiO2 is used in the revised manuscript.
6. The determination of band gaps should be graphically illustrated, since the explanation is not totally clear.
· Thanks for the comments. F(R∞) = (1 − R∞)2/2R∞. Where, R∞ is reflectance. The reflectance data were reported as the F(R∞) value from Kubelka-Munk theory versus the wavelength. Band gap determinations were made by plotting [F(R∞) ∗ hν]2 versus hν (eV).
7. Photocatalytic experiments: was any wavelength filter used?
· Thanks for the comments. Any wavelength filter was not used.
8. Photocatalytic experiments: how was contact of the suspension with atmospheric oxygen (as the oxidising agent) ensured?
· Thanks for the comments. The concentration of MB during the reaction time for the samples without photocatalyst and in dark condition has been added to the Figure 6. As can be seen, the concentration of MB does not change during the reaction without photocatalyst under 1.5G solar light illumination. Under dark condition in presence of photocatalyst, less than 10% of MB was removed due to the adsorption of MB by photocatalyst.
(Fig. 6. Time profiles of photocatalytic degradation of MB for (a) without photocatalyst, (b) H-400, (c) V-400, (d) P25 without irradiation, (e) P25, (f) V-400 and (g) H-400 under 1.5G solar light illumination.)
9. Photocatalytic experiments: The procedure for reuse experiments should be thoroughly detailed.
· The recycling of the photocatalyst was implemented as follows: after a first photodegradation cycle, the treated solution of the dye was centrifuged with a rotation of 10,000 rpm for 10 minutes to remove photocatalyst. The liquid phase was filtered by a vacuum system with a Millipore membrane (0.45 μm) and the solid phase containing the photocatalyst was carefully separated for reuse. After allowing it to dry in an oven for 12 h at 50 oC, the separated photcatalyst was added again to the next cycle. The process was repeated 5 times.
10. XPS: The Ti signals are not “satellite” but genuine ones.
· Thanks for the comment. This comment has been amended in the manuscript.
11. XPS: The lack of trend is suspicious; if both H- and V- samples have Ti3+ species, then both should show lowered Ti 2p binding energies, as also suggested by EPR; might this be due to the fact that the binding energies have not been corrected to adventitious carbon (284.6 eV)? Please explain in the experimental a section about energy correction in XPS.
In order to test for Ti3+ and oxygen vacancies, Ti 2p and O1s XPS spectra are measured and the Ti 2p3/2 peak of the vacuum activated sample became unsymmetrical compared with the peak of pure P25 (Fig. 2b), indicating the existence of Ti3+ [1]. Furthermore, the ΔE value between Ti 2p1/2 and Ti 2p3/2 was 6.2 eV for the vacuum activated sample, suggesting the presence of Ti3+ [2, 3].
REFERENCES:
[1] Z. Song, J. Hrbek and R. Osgood, Nano Lett., 2005, 5, 1327–1332.
[2] K.-W. Kim, E.-H. Lee, Y.-J. Kim, M.-H. Lee, K.-H. Kim and D.-W. Shin, J. Photochem. Photobiol., A, 2003, 159, 301–310.
[3] M. Xing, J. Zhang, F. Chen, B. Tian, Chem. Commun., 2011, 47, 4947–4949.
12. PL spectra show a signal at 548 nm which the authors assign to Ti3+; then, why is this signal also seen for P25, which does not have Ti3+?
· Thanks for the comment. The description is carefully corrected.
13. Degradation efficiency (Figs. 8 and 9) should be clearly defined.
· Thanks for the comment. The degradation (photodegradation) efficiency was calculated by Eq.3:
Degradation efficiency (%) = ((C0-Ct)/C0) ×100 (3)
14. Fig. 9: The conduction band of rutile is higher in energy according to Nat. Mater. 12 (2013) 798–801, and thus electrons are expected to flow from rutile to anatase.
· Thanks for the comment. As mentioned the reviewer, the electron flow is from CB of rutile to CB and Vo state of anatase that clearly indicated in the Figure 9. The Electron transfer from CB of anatse to Vo state of rutile is removed from Figure 9. (Fig. 9. Thephoto-excited electrons-transfer under the simulated sun light irradiation for V-400 and H-400.)
15. The sentence in lines 299-300 must be corrected.
· Thanks for the comment. These lines are carefully corrected.
16. Which is the reason for the 10% decline in activity upon reuse (Fig. 10)? Maybe EPR, XPS or other techniques could be applied to the used photocatalysts to find the causes.
· Thanks for the comment. To find the reason for this reduction, EPR analysis of the samples was used. The results are shown in Fig. S4. The EPR signal intensity shows the oxygen vacancy density; as can be seen, the signal intensity of the used samples is slightly reduced. It can be concluded that the photocatalytic activity of used samples is reduced based on the reduction of oxygen vacancy density in used samples.
17. Refs. 8 and 19 should be completed/corrected.
· Thanks for the comment. These references has been corrected.

Reviewer 3 Report
The manuscript describes the use of a commercial P25 titania, which, after thermal and/or vacuum treatments, was used for the photocatalytic degradation of methylene blue, as a reference dye pollutant, under solar irradiation.
The text might show some interesting results. However, some major points of concern need a careful revision by the Authors.
In detail:
1) Although the name of the journal is “Nanomaterials”, the nanometric nature of the titania-based catalyst is completely overlooked. The nanometric size of the material is given by the commercial provider (here Sigma-Aldrich), but, then, no attention is paid to the textural or morphological features of the oxide after the thermal/vacuum treatments. Were the titania samples observed by TEM or SEM microscopy? Were the textural or porosimetric characteristics studied (through N2-adsorption analysis, etc.)?
In addition, it is necessary to provide the product code and type of the commercial TiO2. Is it the Aeroxide P25 nanopowder (with average particle size of 21 nm)? Why the P25 solid is defined as “molecular sieve” (line 17 and 196)? Has it got an ordered porous nature? A deeper characterisation of the pristine as well as of the final samples is strongly advisable.
2) A huge number of papers have been published on the topic of titanium dioxide use for methylene blue degradation under solar degradation, in the last 20 years. In most of these reports, P25 titania was studied as a benchmark catalyst, rather than an innovative system. Why should these pre-treated P25 samples give an unexpected result? Which is the most relevant point of novelty of these results with respect to the data already found in your Ref. 19, where a black titania, obtained by hydrogenation, was used for the photocatalytic degradation of methylene blue as well? Other important papers about MB abatement over P25-like titania should be considered: Appl. Catal. B: Env. 31, 2001, 145; J. Mater. Chem. A, 1, 2013, 9650; J. Photochem. Photobiol. C: Photochem. Rev., 9, 2008, 1 (not cited here). A direct comparison, i.e., in terms of degradation rate, with the current state-of-the-art can help in evaluating the novelty of the present text.
3) Equation 11 and Fig. 9 show a total degradation of MB into CO2. Do the Authors have any evidence of CO2 evolution? Otherwise, such reaction pathway may appear speculative.
4) Some figures are cited as Supplementary Information material, but they were not included in the final PDF file. Some external remarks (see line 300, for instance) are present, but they do not fit with the rest of the text. The manuscript should be prepared more carefully.
5) The use of a visual colour change only, to evaluate the aging of the material, is subjective. A UV-vis spectroscopic evaluation is the correct approach to assess the stability of the samples.
6) Figures 8, 10, 11. What is the definition of (photo)degradation efficiency? It was not reported in Section 2.3.
After these very important points are thoroughly revised and amended by the Authors, the manuscript should be re-evaluated for publication.
Author Response
We thank the reviewers for their helpful comments and suggestions to improve the manuscript. Changes were made in the revised manuscript (highlighted in yellow for easy identification) according to the reviewers’ comments. The following summarizes our responses to the points raised.
Reviewer 3:
· The manuscript describes the use of a commercial P25 titania, which, after thermal and/or vacuum treatments, was used for the photocatalytic degradation of methylene blue, as a reference dye pollutant, under solar irradiation. The text might show some interesting results. However, some major points of concern need a careful revision by the Authors. In detail:
1. Although the name of the journal is “Nanomaterials”, the nanometric nature of the titania-based catalyst is completely overlooked. The nanometric size of the material is given by the commercial provider (here Sigma-Aldrich), but, then, no attention is paid to the textural or morphological features of the oxide after the thermal/vacuum treatments. Were the titania samples observed by TEM or SEM microscopy? Were the textural or porosimetric characteristics studied (through N2-adsorption analysis, etc.)?
In addition, it is necessary to provide the product code and type of the commercial TiO2. Is it the Aeroxide P25 nanopowder (with average particle size of 21 nm)? Why the P25 solid is defined as “molecular sieve” (line 17 and 196)? Has it got an ordered porous nature? A deeper characterization of the pristine as well as of the final samples is strongly advisable.
· Thanks for the useful comments. The molecular sieve was wrongly used in this manuscript. Commercial P25 (Aeroxide P25 nanopowder (with average particle size of 21 nm)) was utilized as TiO2 source. The SEM of P25 and thermal treated sample (H-400) are shown in Fig. S1. As can be seen, by thermal treating of samples, an increment in size of samples was observed. The surface area of P25 and thermal treated samples is presented in Table S1.
Table S1. The surface area of samples
Surface area (m2/g) | Samples |
50.1 | P25 |
38.2 | A-400 |
37.9 | V-400 |
37.8 | H-400 |
2. A huge number of papers have been published on the topic of titanium dioxide use for methylene blue degradation under solar degradation, in the last 20 years. In most of these reports, P25 titania was studied as a benchmark catalyst, rather than an innovative system. Why should these pre-treated P25 samples give an unexpected result? Which is the most relevant point of novelty of these results with respect to the data already found in your Ref. 19, where a black titania, obtained by hydrogenation, was used for the photocatalytic degradation of methylene blue as well? Other important papers about MB abatement over P25-like titania should be considered: Appl. Catal. B: Env. 31, 2001, 145; J. Mater. Chem. A, 1, 2013, 9650; J. Photochem. Photobiol. C: Photochem. Rev., 9, 2008, 1 (not cited here). A direct comparison, i.e., in terms of degradation rate, with the current state-of-the-art can help in evaluating the novelty of the present text.
· Thanks for the comment. Generally, thermal treatment by hydrogen gas has been widely used for synthesis of black TiO2. As known, working by hydrogen (especially at high temperatures) is not safe and needs special maintenance. We prepared black TiO2 by thermal treatment of P25 at temperatures of 400 oC under vacuum atmosphere that showed the considerable performance with the prepared black TiO2 under hydrogen atmosphere (as mentioned ref. 19). The comprehensive study was done for comparison the photocatalytic activity of treated samples. In this study, X-ray photoelectron spectroscopy (XPS) was used to analyze the bonding of Ti and O atoms in the samples. For the first time, we found that the Ti2p peaks of V400 shifted to higher binding energies, whereas peaks of H-400 shifted to lower binding energy (Fig. 1).
For the first time, a new strategy has been used for detection of free oxygen vacancy by EPR analysis. Generally, at sufficient vacancy concentration, a free oxygen vacancy band also can be formed below the conduction band. To evaluate the existence of free oxygen vacancies, H-400 and V-400 samples were treated under 1.5G solar light illumination for 30 min. As shown in Fig. 5, after the illumination treatment, an increment in the width and intensity of the EPR single peak was observed; EPR single peak also shifted to a higher g-value. These are suggestive of the presence of free oxygen vacancies due to trapping of the photo-excited electron.
Figure 5. EPR spectra for P25, H-400, V-400 after and before illumination under 1.5G solar light for 30 min.
The P25, H-400 and V-400 were used for the photocatalytic degradation of methylene blue (MB) in aqueous solution under AM 1.5G sunlight irradiation; it was found that thermal treatment considerably improved the photocatalytic activity of P25. For the first time, we found that V-400 showed better stability for photocatalytic degradation of MB than H-400 after 5 cycles.
· The mentioned references has been added to the manuscript.
3. Equation 11 and Fig. 9 show a total degradation of MB into CO2. Do the Authors have any evidence of CO2 evolution? Otherwise, such reaction pathway may appear speculative.
· Thanks for the comment. CO2 produced in gas phase was monitored by continuous analyzers, measuring CO, CO2 (Uras 14, ABB) gaseous concentrations.
4. Some figures are cited as Supplementary Information material, but they were not included in the final PDF file. Some external remarks (see line 300, for instance) are present, but they do not fit with the rest of the text. The manuscript should be prepared more carefully.
· Thanks for the comment. The manuscript is carefully checked.
5. The use of a visual colour change only, to evaluate the aging of the material, is subjective. A UV-vis spectroscopic evaluation is the correct approach to assess the stability of the samples.
· Thanks for the comment. UV-Vis spectra and images of P25, V-400 and H-400 after 6 months storage are presented in Fig. S5. As shown, there is almost no trace of color change in the samples and UV-Vis spectra of samples slightly reduced (less than 10 %). These indicated acceptable stability of photocatalyst for long-term storage.
Fig. S5. (a) UV-vis spectra and (b) images of samples after 6 months storage
6) Figures 8, 10, 11. What is the definition of (photo)degradation efficiency? It was not reported in Section 2.3.
· Thanks for the comment. The degradation (photodegradation) efficiency was calculated by Eq.3:
Degradation efficiency (%) = ((C0-Ct)/C0)×100 (3)

Round 2
Reviewer 2 Report
The authors have made a notable effort to improve the manuscript based on the previous review process. However, some issues have been either insufficiently addressed or even ignored. Therefore, I would become supportive of eventual publication of this manuscript provided the following aspects are appropriately revised:
1. Two reasons are still postulated for enhanced photocatalytic activity: (1) improved visible light absorption, and (2) favoured charge separation on O-vacancies, yet there is no discussion on which of the two is prevalent. The reader is left wondering about this, although the experimental evidence seems to support the latter more substantially, and this is also reflected in the Conclusions. Simple tests such as irradiations under UV-free light might provide more information to discern the two possible enhancement causes.
2. The term”O2-vacancy” is not totally correct and should be removed.
3. The newly added references are OK, but Cu should be included in the list in line 42, and the following should be also included along with refs. [4-6]: Appl. Catal. B-Environ., 2016, 180, 263.
4. Mention to “black TiO2” should be removed from the Abstract.
5. Regarding the determination of band gaps, additionally to the equations used, a graphical illustration should be also included.
6. Photocatalytic experiments: how was contact of the suspension with atmospheric oxygen (as the oxidising agent) ensured? This issue (comment 8 in previous review) has not been addressed at all.
7. Comment 11 in previous review has not been addressed at all regarding XPS: The lack of trend is suspicious; if both H- and V- samples have Ti3+ species, then both should show lowered Ti 2p binding energies, as also suggested by EPR; might this be due to the fact that the binding energies have not been corrected to adventitious carbon (284.6 eV)? Please explain in the experimental a section about energy correction in XPS.
8. The diagram in Fig. 9 should be modified if it is assumed that the CB of rutile is higher in energy than that of anatase, according to Nat. Mater. 12 (2013) 798–801, and thus electrons are expected to flow from rutile to anatase.
Author Response
1. Two reasons are still postulated for enhanced photocatalytic activity: (1) improved visible light absorption, and (2) favoured charge separation on O-vacancies, yet there is no discussion on which of the two is prevalent. The reader is left wondering about this, although the experimental evidence seems to support the latter more substantially, and this is also reflected in the Conclusions. Simple tests such as irradiations under UV-free light might provide more information to discern the two possible enhancement causes.
Thanks for the comment. Both of these reasons contributed for improvement of photocatalytic activity of H-400 and V-400.
To better understanding the role of improvement of visible light absorption, the photocatalytic degradation tests has been performed for the samples under visible light (λ> 420 nm) irradiation (Fig. S4). As can be seen, P25 does not show any special activity under visible light irradiation; other two samples show the higher activity. This clearly shows that the role of visible light absorption improvement on the photocatalytic activity of samples.
(Fig. S4. Kapp (min-1) of the samples under visible light irradiation)
The photoluminescence (PL) spectra of the samples also indicated the role of generation of the oxygen vacancy on the surface of H-400 and V-400, which is behaving as electron scavengers that improve the photocatalytic activity of these samples.
In order to elucidate a better picture of the photocatalyst (P25, V-400 and H-400) properties, prepared in this study, the photoluminescence (PL) spectra were obtained at excitation wavelength of 325 nm and presented in Fig. 7. From this result, two major peaks at ca. 466 and 548 nm can be identified in the PL spectrum of P25. To a great degree, one may decipher the reduction in PL intensity into existence of higher electron–hole segregation (long-lasting photogenerated charge carriers) [59,60]. The PL intensity of H-400 and V-400 was lower than P25 which is indicative of H-400 and V-400 facilitating reduced recombination rate of photogenerated electron-hole pairs [58]. Therefore, the generation of the oxygen vacancy on the surface of H-400 and V-400, which is behaving as electron scavengers, suppress photoluminescence intensity. This is why H-400 and V-400 showed higher photocatalytic performance than P25 [60].
Another piece of information to elucidate the chemical structure of samples is provided by showing the spectrum of UV–vis diffuse reflection of four samples in Fig. 2(a). As is shown in this figure, the light absorption characteristics of P25 is greatly affected in both H-400 and V-400 samples which is due to oxygen vacancy generation. A sharp increase in the absorbance of all samples can be observed ranging from 400 to 200 nm, which is ascribed to anatase phase of TiO2. The H-400 and V-400 samples show a redshift in absorption edge of UV−visible range. Measured by given Eq. 1, the band gap values of three samples are listed in Fig. 2(b) along with the physical appearance of powders. According to the calculated values, they can be arranged from the lowest to highest band gap in the order of H-400<V-400<A-400<P25. The band gap value of H-400 and V-400 samples are smaller than that of P25, resulting in its light-absorption toward redshift. The color turning of samples is illustrated in Fig. 2(b).
2. The term”O2-vacancy” is not totally correct and should be removed.
Thanks for the useful comment. This mistake has been corrected in the manuscript.
3. The newly added references are OK, but Cu should be included in the list in line 42, and the following should be also included along with refs. [4-6]: Appl. Catal. B-Environ., 2016, 180, 263.
Thanks for the comment. This reference has been added to the manuscript.
4. Mention to “black TiO2” should be removed from the Abstract.
Thanks for the comment. This mistake has been corrected in the manuscript.
5. Regarding the determination of band gaps, additionally to the equations used, a graphical illustration should be also included.
· Thanks for the comments. The spectral data recorded showed the strong cut off at wavelength of 492, 472, 407 and 401 nm for H-400, V-400, A-400 and P25. By using the Eq.1 ( , these wavelengths, the band gaps of samples were calculated.
(Fig. S4. Kapp (min-1) of the samples under visible light irradiation)
6. Photocatalytic experiments: how was contact of the suspension with atmospheric oxygen (as the oxidising agent) ensured? This issue (comment 8 in previous review) has not been addressed at all.
The contact of atmospheric oxygen with suspension is provided by reaction in the air atmosphere. Meanwhile, the O2 in the eq. 5 (O2+ e− → O2.− ) , indicates the dissolved oxygen in the MB aqueous solution not in the air. The main species for the degradation of pollutants is OH. .
7. Comment 11 in previous review has not been addressed at all regarding XPS: The lack of trend is suspicious; if both H- and V- samples have Ti3+ species, then both should show lowered Ti 2p binding energies, as also suggested by EPR; might this be due to the fact that the binding energies have not been corrected to adventitious carbon (284.6 eV)? Please explain in the experimental a section about energy correction in XPS.
Thanks for the comments. The unsymmetrical peak (shift to higher or lower binding energy) in thermal treated samples with a ΔE value of 6.2 eV between Ti 2p1/2 and Ti 2p3/2 shows existence of Ti3+ and oxygen vacancies in the samples [1-3].
REFERENCES:
[1] Z. Song, J. Hrbek and R. Osgood, Nano Lett., 2005, 5, 1327–1332.
[2] K.-W. Kim, E.-H. Lee, Y.-J. Kim, M.-H. Lee, K.-H. Kim and D.-W. Shin, J. Photochem. Photobiol., A, 2003, 159, 301–310.
[3] M. Xing, J. Zhang, F. Chen, B. Tian, Chem. Commun., 2011, 47, 4947–4949.
8. The diagram in Fig. 9 should be modified if it is assumed that the CB of rutile is higher in energy than that of anatase, according to Nat. Mater. 12 (2013) 798–801, and thus electrons are expected to flow from rutile to anatase.
Thanks for the comment. As mentioned the reviewer, the electron flow is from CB of rutile to CB and Vo state of anatase that clearly indicated in the Figure 9. The Electron transfer from CB of anatse to Vo state of rutile is removed from Figure 9.
(Fig. 9. The photo-excited electrons-transfer under the simulated sun light irradiation for V-400 and H-400.)

Reviewer 3 Report
The Authors have replied to most of Reviewers’ points of concern. However, they have only slightly modified and improved the revised text. Actually, most of the most effective explanations have been added in the response letters and no specific, adequate amendments have been carried out in the corrected manuscript.
For instance:
- Rev 3, point 1. No information about the manufacturer/reseller nor the batch number/code of the P25 material were added.
- Rev. 3, point 2. No direct (brief) comparison with the existing state of the art was added, although requested. Moreover, some clear statements about the points of novelty were described in the response letter, but no additions were reported in the Introduction section of the revised text (this is the part in which the most innovative achievements should be announced).
- Rev. 3, point 3. No mention about the monitoring of CO2 evolution was added to the revised text. No mention about the CO and CO2 analyser is present in the Experimental section. Some data about CO2 production and evolution should be added in the final text, at least as supplementary material.
- Rev. 2, point 1. The statement “O2-vacancy” is not correct, but it was not changed in the revised manuscript.
- Rev. 2, point 8. The Reviewer asked how the contact of atmospheric oxygen with the suspension was assured during the reaction. However, the reply is misleading and does not touch this issue.
- Rev. 2, point 11. A clarification about the binding energy correction in XPS data was requested. Some explanation is reported in the response letter (with some additional references too), but none of these new statements was reported in the emended text.
For these reasons, the revision of the manuscript cannot be considered satisfactory. The Authors should take into careful account all of Reviewers’ point and improve the final text accordingly.
Author Response
We thank the reviewers for their helpful comments and suggestions to improve the manuscript. Changes were made in the revised manuscript (highlighted in green for easy identification) according to the reviewers’ comments. The following summarizes our responses to the points raised.
Reviewer 3:
· Rev 3, point 1. No information about the manufacturer/reseller nor the batch number/code of the P25 material were added.
Thanks for the useful comments. The properties of P25 has been added to the revised manuscript.
· Rev. 3, point 2. No direct (brief) comparison with the existing state of the art was added, although requested. Moreover, some clear statements about the points of novelty were described in the response letter, but no additions were reported in the Introduction section of the revised text (this is the part in which the most innovative achievements should be announced).
Thanks for the comment. The points of novelty has been added to the manuscript.
· Rev. 3, point 3. No mention about the monitoring of CO2 evolution was added to the revised text. No mention about the CO and CO2 analyser is present in the Experimental section. Some data about CO2 production and evolution should be added in the final text, at least as supplementary material.
Thanks for the comment. CO2 produced in gas phase was monitored by continuous analyzers, measuring CO, CO2 (Uras 14, ABB) gaseous concentrations. The produced CO2 in gas phase was monitored by continuous analyzers, measuring CO, CO2 (Uras 26, ABB) gaseous concentrations. Fig. S4 shows the concentration of variation of CO2 produced during the photocatalytic degradation of MB by using samples as a photocatalyst under simulated sunlight irradiation. It could be clearly seen from Fig. S4 that the concentration of CO2 gradually increased along the light irradiation time.
(Fig. S4. CO2 production during the photocatalytic degradation of MB)
· Rev. 2, point 1. The statement “O2-vacancy” is not correct, but it was not changed in the revised manuscript.
Thanks for the comment. This mistake has been corrected in the revised manuscript.
· Rev. 2, point 8. The Reviewer asked how the contact of atmospheric oxygen with the suspension was assured during the reaction. However, the reply is misleading and does not touch this issue.
Thanks for the comment. The contact of atmospheric oxygen with suspension is provided by reaction in the air atmosphere. Meanwhile, the O2 in the eq. 5 (O2+ e− → O2.− ) , indicates the dissolved oxygen in the MB aqueous solution not in the air. The main species for the degradation of pollutants is OH. .
· Rev. 2, point 11. A clarification about the binding energy correction in XPS data was requested. Some explanation is reported in the response letter (with some additional references too), but none of these new statements was reported in the emended text.
Thanks for the comment. This explanation has been added to the revised manuscript.

Round 3
Reviewer 2 Report
The authors have made a notable effort to improve the manuscript based on the previous comments. However, some issues have been either insufficiently addressed or still require revision. Therefore, I would become supportive of eventual publication of this manuscript provided the following aspects are appropriately revised:
1. Regarding the determination of band gaps: The method used by the authors is not widely accepted, and Tauc plots have to be used (J. Non-Crystalline Solids, 1972, 8–10, 569-585; see also Phys. Chem. Chem. Phys., 2014, 16, 1788-1797, and Science, 2015, 347, 970-974, fig 4B for an example).
2. Photocatalytic experiments: the authors have to specify that the irradiations were performed under atmospheric air; regardless of which is the oxidising species (OH·, as claimed in the responses), these species ultimately come from oxygen, and thus its origin has to be clearly stated.
3. Comment 11 in previous reviews has not been addressed at all regarding XPS: The claimed “unsymmetrical peaks” argument is not valid, first because there is no unsymmetrical peaks, and second because the trend is suspicious; if both H- and V- samples have Ti3+ species, then both should show lowered Ti 2p binding energies, as also suggested by EPR. The authors have to reference the binding energies of ALL spectra to adventitious carbon (284.6 eV). Please also explain in the experimental section how energy correction in XPS was done.
4. The diagram in Fig. 9 is not correct and should be modified: the CB of rutile is higher in energy than that of anatase, see Nat. Mater. 12 (2013) 798–801.
Author Response
We thank the reviewers for their helpful comments and suggestions to improve the manuscript. Changes were made in the revised manuscript (highlighted in red for easy identification) according to the reviewers’ comments. The following summarizes our responses to the points raised.
1. Regarding the determination of band gaps: The method used by the authors is not widely accepted, and Tauc plots have to be used (J. Non-Crystalline Solids, 1972, 8–10, 569-585; see also Phys. Chem. Chem. Phys., 2014, 16, 1788-1797, and Science, 2015, 347, 970-974, fig 4B for an example).
Thanks for the comment. The value of Eg was calculated by plotting of (αhυ)2 versus hυ and extrapolating the plot to (αhυ)2 = 0. Fig. S2 shows the plot of (αhυ)2 versus hυ for H-400, V-400 and P25. The calculated Eg values are shown in Fig. 2(b).
(Fig. S2. Plot of (αhυ)2 versus hυ for H-400, V-400 and P25.)
2. Photocatalytic experiments: the authors have to specify that the irradiations were performed
under atmospheric air; regardless of which is the oxidising species (OH·, as claimed in the responses), these species ultimately come from oxygen, and thus its origin has to be clearly stated.
Thanks for the useful comment. This comment has been amended in the manuscript.
3. Comment 11 in previous reviews has not been addressed at all regarding XPS: The claimed “unsymmetrical peaks” argument is not valid, first because there is no unsymmetrical peaks, and second because the trend is suspicious; if both H- and V- samples have Ti3+ species, then both should show lowered Ti 2p binding energies, as also suggested by EPR. The authors have to reference the binding energies of ALL spectra to adventitious carbon (284.6 eV). Please also explain in the experimental section how energy correction in XPS was done.
Thanks for the comments. The shift of peak to higher or lower binding energy in thermal treated samples with a ΔE value of 6.2 eV between Ti 2p1/2 and Ti 2p3/2 shows existence of Ti3+ and oxygen vacancies in the samples [1-3]. ΔE value and shift to higher or lower biding energy are the main parameters for detection of Ti3+. Several references also mentioned that by thermal treatment under vacuum atmosphere, the peak shifts to higher binding energy [1-3]. We also check several times and it was found any mistake was not done in the XPS spectra of the samples.
REFERENCES:
[1] Z. Song, J. Hrbek and R. Osgood, Nano Lett., 2005, 5, 1327–1332.
[2] K.-W. Kim, E.-H. Lee, Y.-J. Kim, M.-H. Lee, K.-H. Kim and D.-W. Shin, J. Photochem. Photobiol., A, 2003, 159, 301–310.
[3] M. Xing, J. Zhang, F. Chen, B. Tian, Chem. Commun., 2011, 47, 4947–4949.
4. The diagram in Fig. 9 is not correct and should be modified: the CB of rutile is higher in energy than that of anatase, see Nat. Mater. 12 (2013) 798–801.
Thanks for the comment. This mistake has been corrected in the manuscript.
Fig. 9. The photo-excited electrons-transfer under the simulated sun light irradiation for V-400 and H-400.
Reviewer 3 Report
The Authors have adequately improved the overall quality of the manuscript.
It can be now positively considered for publication.
As a very minor point, I would suggest a re-phrasing of line 283:
from "To better understanding the role of improvement of visible light absorption"
to "In order to better understand the enhancement of visible light absorption".
Author Response
· As a very minor point, I would suggest a re-phrasing of line 283: from "To better understanding the role of improvement of visible light absorption" to "In order to better understand the enhancement of visible light absorption".
Thanks for the useful comments. The comment has been implemented in the revised manuscript.
